# Chemical and entropic control on the molecular self-assembly process

Daniel M. Packwood[1,2,3], Patrick Han[1,4] & Taro Hitosugi[1,5]

Molecular self-assembly refers to the spontaneous assembly of molecules into larger structures. In order to exploit molecular self-assembly for the bottom-up synthesis of nanomaterials, the effects of chemical control (strength of the directionality in the intermolecular interaction) and entropic control (temperature) on the self-assembly process should be clarified. Here we present a theoretical methodology that unambiguously distinguishes the effects of chemical and entropic control on the self-assembly of molecules adsorbed to metal surfaces. While chemical control simply increases the formation probability of ordered structures, entropic control induces a variety of effects. These effects range from fine structure modulation of ordered structures, through to degrading large, amorphous structures into short, chain-shaped structures. Counterintuitively, the latter effect shows that entropic control can improve molecular ordering. By identifying appropriate levels of chemical and entropic control, our methodology can, therefore, identify strategies for optimizing the yield of desired nanostructures from the molecular self-assembly process.

[1] Advanced Institute for Materials Research (WPI-AIMR), Tohoku University, Sendai 980-8577, Japan. [2] Institute for Integrated Cell-Material Sciences (WPI-iCeMS), Kyoto University, Kyoto 606-8501, Japan. [3] Japan Science and Technology Agency (PRESTO), Kawaguchi, Saitama 332-0012, Japan. [4] California NanoSystems Institute and Departments of Chemistry and Biochemistry and Materials Science and Engineering, University of California, Los Angeles, Los Angeles, California 90095, USA. [5] School of Materials and Chemical Technology, Tokyo Institute of Technology, Tokyo 152-8352, Japan. Correspondence and requests for materials should be addressed to D.M.P. (email: dpackwood@icems.kyoto-u.ac.jp).

Molecular self-assembly, which refers to the spontaneous assembly of precursor molecules to form nanostructured objects[1], is controlled by the intrinsic properties of the molecules and their environment. In order to tailor the structures that emerge from the self-assembly process, we must rely on the following indirect strategy: tuning the directionality of the intermolecular interaction by design of the precursor molecule structure, and careful choice of the temperature (Fig. 1). The strength of the interaction directionality and the height of the temperature can be referred to as the chemical control and entropic control, respectively, under which self-assembly occurs. If the entropic control is very weak, then the self-assembly process is under chemical control and molecules assemble according to the directionality of the molecule–molecule interaction (A). If the entropic control is much stronger than the chemical control, then assembly does not occur and the precursor molecules remain randomly dispersed across the medium. However, when the entropic control is neither weak nor strong compared with the chemical control, it is not possible to guess what kinds of structures will be produced by the molecular self-assembly process. Full characterization of chemical and entropic controls is vital for molecular self-assembly to be used for systematic fabrication of precise nanomaterials.

Chemical and entropic controls are ultimately related to the thermodynamics of self-assembly process and are best characterized *via* theoretical studies. However, several problems are encountered when applying atomistic computational approaches to molecular self-assembly phenomena. Molecular self-assembly takes place over enormous, often microsecond-exceeding, time scales, making the prediction of thermodynamically stable molecular assemblies with atomistic models prohibitive. While some remarkable progress has been made in this area[2–11], there is little consensus in the literature on how molecular self-assembly should be simulated. The lack of molecule-surface force fields for the important case of molecular self-assembly on metal surfaces further limits the feasibility of atomistic simulation, although promising progress is being made here as well[12–15]. An arguably more serious issue is that atomic simulations do not directly address the effects of chemical and entropic controls on the molecular self-assembly process. Instead, they yield large volumes of data that require lengthy post-simulation analysis, and it is not clear what kind of analysis is needed for the study of chemical and entropic controls. In order to surmount these difficulties, it is necessary to develop novel computational techniques that unambiguously separate the effects of chemical and entropic controls on molecular self-assembly without difficult post-simulation analysis.

In this paper, we present a theoretical methodology for molecular self-assembly on metal surfaces that unambiguously distinguishes the effects of entropic control and chemical control under low surface coverage conditions. In order of decreasing chemical control (decreasing interaction directionality), we study the self-assembly of $10,10'$-dibromo-$9,9'$-bianthracene (Br$_2$BA) (refs 16,17), $10,10'$-diamine-$9,9'$-bianthracene ((NH$_2$)$_2$BA), and $10,10'$-dimethyl-$9,9'$-bianthracene (Me$_2$BA) molecules on copper (111) surfaces (Fig. 1b). We find that, while

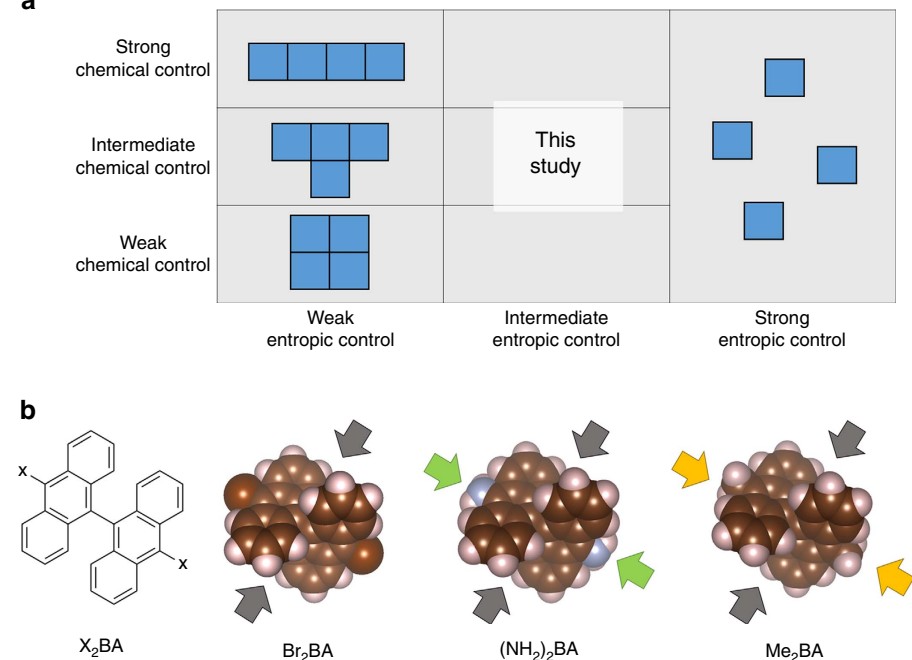

**Figure 1 | Chemical and entropic controls in molecular self-assembly.** (**a**) Effect of chemical and entropic controls on the structures formed by molecular self-assembly. Chemical control refers to the strength of the interaction directionality between molecules, and entropic control refers to temperature. The blue blocks represent molecules adsorbed to a solid substrate, viewed with the substrate in the plane of the page. (**b**) $10,10'$-dibromo-$9,9'$-bianthracene (Br$_2$BA), $10,10'$-diamine-$9,9'$-bianthracene ((NH$_2$)$_2$BA), and $10,10'$-dimethyl-$9,9'$-bianthracene (Me$_2$BA) molecules in their adsorption conformations on a copper(111) surface, viewed with the surface in the plane of the page. Br$_2$BA molecules represent strong chemical control, (NH$_2$)$_2$BA molecules represent intermediate chemical control, and Me$_2$BA molecules represent weak chemical control. Grey, green, and orange arrows represent bianthryl tips, amine groups, and methyl groups, respectively. Br$_2$BA molecules have a strong tendency to interact with each other *via* their anthryl tips, giving the molecules a strong interaction directionality. (NH$_2$)$_2$BA molecules can interact with each other *via* hydrogen bonding between amine groups, reducing the preference for interactions in the bianthryl tip direction compared with Br$_2$BA. In Me$_2$BA molecules, σ-conjugation between the C–H bonds of the methyl groups and the *p*-orbitals of the anthryl units spreads the π-system onto the methyl groups, allowing for methyl–methyl and also methyl–anthryl interactions between molecules and reducing the interaction directionality compared with both Br$_2$BA and (NH$_2$)$_2$BA.

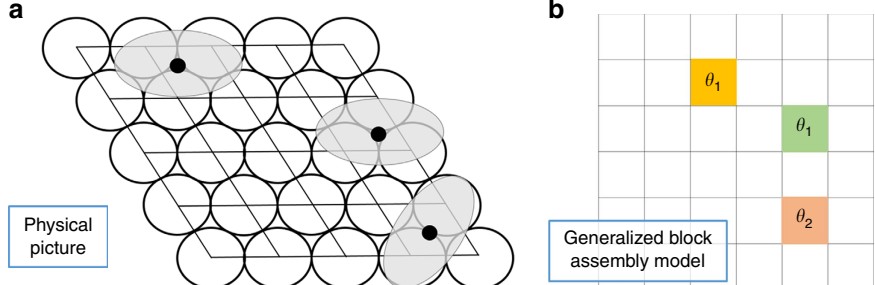

**Figure 2 | Generalized block assembly model.** (**a**) Metal(111) surface with unit cells highlighted. The ovals represent molecules adsorbed to the surface. The adsorption site of the molecules lie directly beneath the center of mass of the molecule, which is indicated by the black spots. (**b**) The configuration of the generalized block assembly model corresponding to the adsorption pattern in **a**. The grid corresponds to the unit cells of the 111 surface and the coloured cells correspond to unit cells which carry an adsorbed molecule. The colour indicates the adsorption site for the molecule, and the symbols $\theta_1, \theta_2, \ldots$ correspond to an orientation for the molecule.

decreasing chemical control simply decreases the assembly probability of ordered, chain-type structures compared with disordered, amorphous structures, entropic control has two main effects: to degrade amorphous structures into chain-type structures, and to generate asymmetry in the adsorption sites of molecules within chain-shaped structures. Interestingly, the first effect shows that entropic controls can be used to improve the overall structural quality of assemblies formed from the molecular self-assembly process. Conversely, the second effect shows that entropic controls also reduce structural quality on a fine scale. Moreover, we show that the formation of amorphous structures is an effect of weak chemical control rather than strong entropic control. These insights provide much-needed strategies for the fabrication of nanomaterials on metal surfaces.

## Results

**Computational approach**. Our computational approach, which we refer to as GAMMA modelling ( = Generalized block AsseMbly Machine learning equivalence clAss sampling modelling), has three components. (1) A general, Ising-type model for adsorbed molecules on a metal surface (the generalized block assembly (GBA) model). (2) A molecule-surface and molecule–molecule interaction energy function constructed by machine learning of density functional theory (DFT)-derived data. (3) Equivalence class sampling (ECS), which directly incorporates free energy into the usual Monte Carlo framework and deals away with the long-time scales of the self-assembly phenomenon. Thus, (1) utilizes an approach from the 'coarse-grained' end of the computational spectrum, (2) connects our approach with the 'atomistic' end of the spectrum, and (3) allows for efficient elucidation of the stable assemblies. GAMMA modelling is specifically intended to study molecular self-assembly under vacuum conditions, and is not applicable to self-assembly that occurs in the liquid phase.

Figure 2 describes component (1) of GAMMA modelling, namely the GBA model. This model considers a grid of cells, labelled as $C_1, \ldots, C_m$, which correspond to the unit cells of the surface. Each cell possesses $k$ possible colours, labelled as $\sigma_1, \ldots, \sigma_m$, where each colour corresponds to a possible adsorption site for the molecule. Moreover, colour $\sigma_j$ possesses $q_j$ shades, $\theta_{j,1}, \ldots, \theta_{j,q_j}$. These shades correspond to the possible orientations that may be adopted by a molecule residing at an adsorption site corresponding to colour $\sigma_j$. An individual molecule $z$ corresponds to a single cell $C_i$ and a colour-shade combination $(\sigma_j, \theta_{j,h})$. A configuration is any choice of $N$ molecules, where $N$ is fixed. The energy of a configuration $c$ is

given by the energy function

$$E(c) = \sum_{z \in c} v(z) + \frac{1}{2} \sum_{z_i, z_j \in c} u(z_i, z_j) \qquad (1)$$

where the first sum runs over all molecules and the second sum runs over all pairs of molecules. The term $v(z)$ measures the adsorption energy of the molecule to the surface, and $u(z_i, z_j)$ measures the interaction energy between molecules $z_i$ and $z_j$. equation (1) assumes that a 'surface-assisted molecular self-assembly' is operating, in which the adsorption sites and orientations for the molecules are determined by the surface–molecule interaction rather than the intermolecular interactions. This assumption has been shown to apply to bianthracene molecules adsorbed to metal surfaces[16,17].

Component (2) of GAMMA modelling relates to the construction of the energy function in equation (1). DFT calculations identified nine adsorption sites per unit cell for dibromo-bianthracene molecules adsorbed to Cu(111), each permitting one or two stable orientations for the molecule. These stable adsorption sites and orientations correspond to the colour-shade combinations that were used in the model (see the Methods section, Supplementary Notes 1 and 2, Supplementary Figs 1–6, and Supplementary Tables 1 and 2). For each of these colour-shade combinations, $v(z)$ in equation (1) was calculated directly *via* DFT. $v(z)$ typically has values between $-1.8$ to $-2.2$ eV. This strong surface–molecule interaction justifies the major assumptions in equation (1) (see Supplementary Note 3, Supplementary Figs 7–9, Supplementary Tables 3–5). As for the molecule–molecule interaction function $u(z_i, z_j)$, note that if we allow for 2,500 cells in the model, then there are $\sim 3 \times 10^8$ possible pairwise interactions $(z_i, z_j)$ to consider. With the computational resources available to us, it would take over 4,000 years to calculate $u$ for every possible pairwise interaction. We, therefore, employed a machine learning approach, in which a random sample of 4,000 to 5,000 pairs of molecules was generated, with each molecule in the fixed conformation as described above. The interaction energy for each pairwise interaction was calculated *via* DFT, again with the molecular conformation kept fixed. Then, we fit a function $\hat{u}$ that approximates the true (unknown) interaction energy function $u$. $\hat{u}$ was validated against test data obtained by DFT (see Supplementary Note 4 and Supplementary Figs 10–14 for a description of the fitting procedure). Coulomb matrices were used to represent pairwise interactions[18,19]. The interaction energies ranged between about $-0.25$ to $-0.05$ eV for attractive ($u < 0$) interactions.

Figure 3 describes component (3) of GAMMA modelling, namely ECS[20] (see Supplementary Note 5 and Supplementary

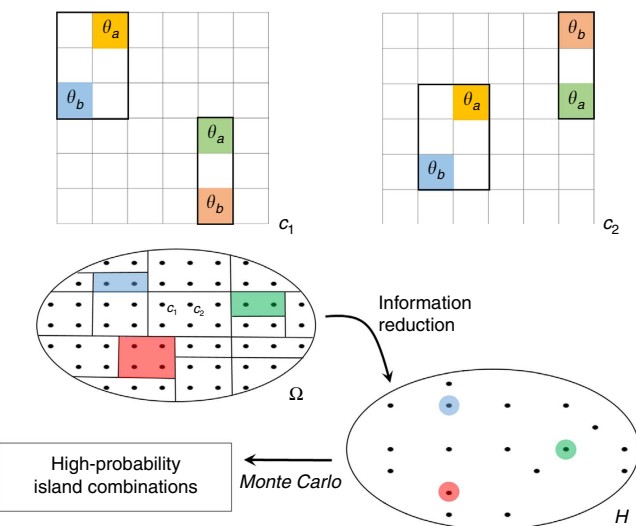

**Figure 3 | Identification of high-probability island combinations via equivalence class sampling.** $c_1$ and $c_2$ are two possible configurations of the generalized block assembly model. Islands are groups of molecules contained in the black boxes. $\Omega$ represents the configuration space, in which each black points represents a single configuration. Configuration $c_1$ can be transformed into configuration $c_2$ by twofold rotation and translation of islands about the grid. Under this transformation, the configuration space $\Omega$ is partitioned into so-called equivalence classes (indicated by dividing lines inside of $\Omega$). A reduced configuration space $H$ is created by considering only the unique equivalence classes. The points inside of $H$ represent entire equivalence classes, and can be interpreted as possible island combinations in the GBA model. Monte Carlo sampling from $H$ then identifies the high-probability island combinations.

Figs 15–22). The fundamental principle of ECS is the elimination of superfluous information from the configuration space $\Omega$. As well as dramatically improving Monte Carlo convergence times, this information reduction procedure embeds a configurational entropy term (and hence configurational free energy) into the Monte Carlo framework. To carry out the information reduction procedure, we define an island as a group of molecules in which no molecule separated from the others in the group by a distance greater than $M_c$, where $M_c$ is an intermolecular interaction cut-off (see the Methods section). Two configurations $c_1$ and $c_2$ are said to belong to the same class if their islands can be superimposed by two-fold rotation and translation. A reduced configuration space $H$ is then obtained by considering only the unique classes of configurations. The space $H$ only contains information on the combinations of islands that can occur, and does not contain any information on the position or orientation of the islands on the surface. The latter information is irrelevant under conditions of low surface coverage, in which the interaction between molecular islands is insignificant. Following this information reduction procedure, the probability that class $q$ (or island combination $q$) occurs at equilibrium works out to be

$$\nu(q) = \frac{1}{Q} \exp\left(\frac{-A(q)}{k_B T}\right), \tag{2}$$

where $Q$ is a normalizer constant, $T$ is the thermodynamic temperature and $A(q)$ is the configurational free energy of island combination $q$. Explicitly, we have $A(q) = E(c; q) - TS(q)$, where $E(c; q)$ is the energy for any configuration $c$ that contains island combination $q$ (given by equation (1)), and

$$S(q) = k_B \ln n(q) \tag{3}$$

is a configurational entropy arising from the number of ways $n(q)$ of arranging the island combination $q$ on the surface. ECS then involves sampling island combinations from the reduced configuration space $H$ via Monte Carlo sampling. Because GAMMA modelling employs frozen molecular and surface geometries, it excludes conformational contributions to the entropy such as dissipation of heat into the metal lattice or into the internal modes of the molecules. However, the molecular self-assembly phenomenon occurs on the scale of molecule configurations, and, therefore, effects that occur on the scale of molecule conformations are not expected to be relevant to this study.

The most important feature of our formalism is that the effect of entropic control on the molecular self-assembly process in the intermediate regime can be completely characterized by inspection of the formula for $n(q)$. For simplicity, we consider the special case where $q$ contains two islands $I_1$ and $I_2$. For now, we fix the orientations $o_1$ and $o_2$ for islands $I_1$ and $I_2$, respectively. Under conditions of low surface coverage, in which the surface area occupied by islands is negligible, we can calculate $n(q)$ by 'shrinking' the islands so that they occupy a single unit cell (Supplementary Note 5). Then, the number of ways to arrange $I_1$ and $I_2$ on the surface in this fixed orientation under low coverage conditions is the simply number of ways to choose two cells from the grid, that is,

$$n(q; o_1, o_2)^* = \frac{d!}{(d-2)!\,2!} \tag{4}$$

where $d$ is number of cells in the model. Actually, equation (4) is only correct when the islands $I_1$ and $I_2$ are indistinguishable upon interchange, that is, $I_1$ can be superimposed onto $I_2$ by translating it across the grid. If $I_1$ and $I_2$ are distinguishable upon interchange, then equation (4) under-counts the number of ways to arrange the islands on the surface. Equation (4) should then be replaced with the formula

$$n(q; o_1, o_2) = \frac{2!}{A_1!\,A_2!} n(q; o_1, o_2)^*. \tag{5}$$

Here, $A_1 = 2$ and $A_2 = 0$ if $I_1$ and $I_2$ are indistinguishable upon interchange, whereas we have $A_1 = 1$ and $A_2 = 1$ otherwise. Equation (5) holds when $I_1$ and $I_2$ are in fixed orientations $o_1$ and $o_2$, respectively. The total number of ways to arrange the islands $I_1$ and $I_2$ on the surface is then obtained by summing (5) over all island orientations, that is,

$$n(q) = n(q; o_1, o_2) + n(q; o_1', o_2) + n(q; o_1, o_2') + n(q; o_1', o_2'), \tag{6}$$

where $o_k'$ is the other orientation available to island $I_k$ ($k = 1$ or 2). The orientation $o_k'$ is obtained by rotating island $I_k$ in orientation $o_k$ by 180°. If $I_k$ possesses twofold rotational symmetry, then the terms in equation (6) involving $o_k'$ will not be present. Therefore, $n(q)$ (and hence the entropy $S(q)$) is large when the islands do not possess twofold symmetry or are indistinguishable under twofold rotations and translations; in the former case more terms will appear in equation (6), whereas in the latter case the terms in equation (6) will be large because the factors in the denominator of equation (5) will be equal to 1. Equations (5 and 6) have essentially the same form for the case where the number of islands is not equal to 2. In the general case of more than two islands, we also find that $S(q)$ tends to grow as the number of islands increases. As entropic control become stronger and the contribution of the entropy $S(q)$ to the free energy increases, the formation of large numbers of rotationally asymmetric, distinguishable islands becomes thermodynamically favourable. This general result characterizes entropic control regardless of the type

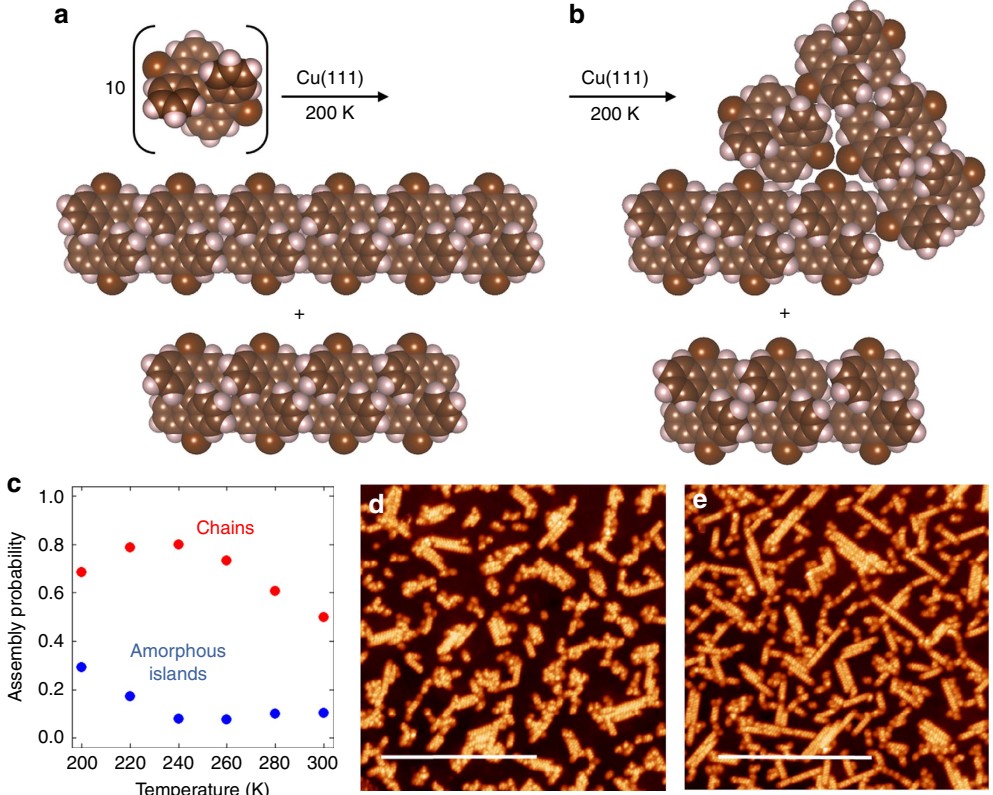

**Figure 4 | Self-assembly under strong chemical control.** (**a**,**b**) Two typical, low free-energy island combination resulting self-assembly of $Br_2BA$ molecules on Cu(111). In **a** both islands have a chain shape. In **b** the top island has an amorphous shape. (**c**) Probabilities for assembly of chains and amorphous islands. (**d**,**e**) STM (scanning tunnelling microscope) image of $Br_2BA$ islands on Cu(111). The scale bars correspond to 43 nm. STM conditions: Sample bias voltage = 1.1 V, tunnelling current = 10 pA, STM imaging temperature = 5.6 K, annealing temperature = 360 °C (**d**) and 400 °C (**e**). (**d**) Shows mainly amorphous islands, whereas (**e**) shows mainly chain-shaped islands.

of molecule under consideration. By running simulations for different kinds of molecules and looking for effects exterior to the above, we can, therefore, unambiguously characterize the effect chemical control on the molecular self-assembly process.

In order to unambiguously characterize the effects of entropy on the molecular self-assembly process, it is necessary for simple, analytic formulas for the entropy (such as equations (4–6)) to be available. The reason why we restrict ourselves to low-coverage conditions is because it appears impossible to obtain closed-form expressions for the degeneracy factors under higher coverage conditions. More general formulas for the entropy are available in the literature (for example, ref. 21), and they could be used in place for formula (4) without any other alterations to the theory. However, these formulas are not in closed form and, therefore, do not permit a straightforward and unambiguous interpretation.

Note that the image in Fig. 2b should be treated as shorthand notation for the real space configuration of the molecules on the surface. In particular, the apparent fourfold symmetry of the GBA model in Fig. 2b is never imposed during our calculations. Given a set of molecules (cell, colour and shade combinations in the GBA model), we always map back to the real three-dimensional space where the surface and molecule atoms lie and perform our energy calculations there. The 'grid' on which our calculations are performed consists of the actual adsorption sites on the surface (Supplementary Note 2 and Supplementary Fig. 6), which is determined directly from the surface–molecule interaction potential. The entropic component is calculated by first condensing each unit cell into a single point, and then calculating $n(q)$ by considering the number of ways

of choosing distinct sets of points from this lattice. This is a typical combinatorial calculation and does not depend upon the symmetry of the lattice. Our calculation, therefore, do not suffer from any artificial fourfold symmetries.

**Molecular self-assembly under strong chemical control.** Chemical control cannot be unambiguously measured, however, it can be qualitatively regarded as the tendency for two molecules to align in the direction of their bianthryl tips in the limit of 0 K, at equilibrium. Chemical control will be large for $Br_2BA$ (since the Br substituent interacts only weakly with the Br atoms and bianthryl tips of other molecules), of intermediate size for $(NH_2)_2BA$ (since the N atom can form hydrogen bonds with the H atoms of other $NH_2$ groups), and small for $Me_2BA$ (since the anthracene $p$ electrons can feed into the CH bond of $CH_3$ via $\sigma$-conjugation, removing the specificity for $\pi$–$\pi$ interactions in the direction of the bianthryl tips). We assume that chemical control can be expressed in units of energy. We first consider the case of $Br_2BA$ self-assembly on Cu(111), in which the chemical control is the strongest of all cases studied here. Figure 4a presents typical, low free energy island combinations for this case at low temperatures (200 K). At this temperature, we estimate a roughly 70% chance of forming chain-shaped islands and a 30% chance of forming amorphous islands (islands which lack a clear chain shape). These amorphous islands typically consist of multiple small chains closely packed together, and are expected to absent when chemical control is extremely strong. As we strengthen the entropic control, the probability of forming amorphous islands decreases.

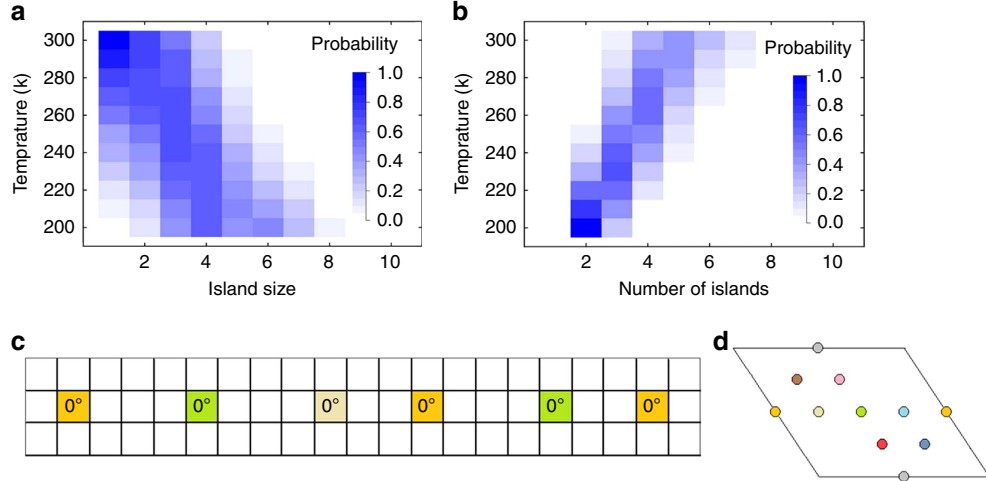

**Figure 5 | Island statistics and fine-structure under strong chemical control.** (**a,b**) Distribution of island sizes (number of molecules in an island) and number of islands at various temperatures for the Br$_2$BA simulations. (**c**) Model representation of the top chain from Fig. 4a. (**d**) Correspondence between the model colours and adsorption site locations in the Cu(111) unit cell. By convention, an orange or grey coloured cell in the model corresponds to the adsorption sites on the far right and far bottom of the Cu(111) unit cell, respectively.

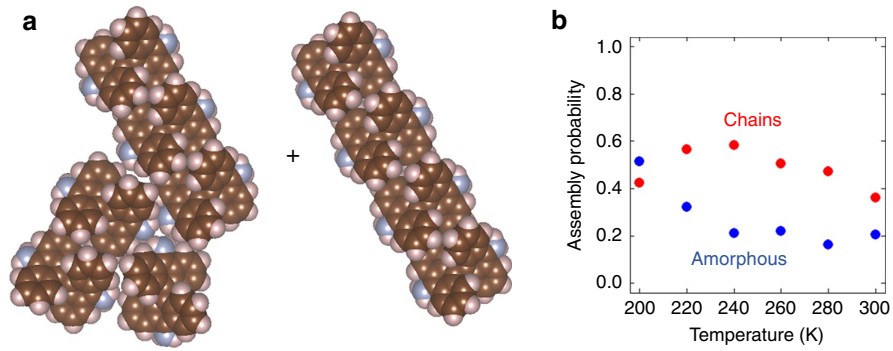

**Figure 6 | Self-assembly under intermediate chemical control.** (**a**) A typical low-free energy island combination formed by self-assembly of (NH$_2$)$_2$BA at 200 K on Cu(111). The island on the left has an amorphous shape, and the island on the right has a chain shape. (**b**) Assembly probabilities for chain-shaped and amorphous islands.

On the other hand, the probability of forming chains increases to a maximum and then decays as entropic controls are enforced. Scanning tunnelling microscopy (STM) data in Fig. 4d,e indeed suggest the presence of amorphous islands of Br$_2$BA molecules on Cu(111) at low temperatures, and an increased fraction of chains following a comparable (~40 K) temperature jump. Note that the STM images were collected under higher surface coverages than those considered in our calculations, and, therefore, the temperature range over which self-assembly occurs differs between theory and experiment. In Supplementary Note 5, it is shown that the predictions of our theory will still be relevant for slight deviations from low-coverage conditions, which accounts for the fact that our theory can explain some aspects of the STM data.

The results in Fig. 4 can be understood by considering Fig. 5a,b, which show that as the temperature increases the low-free energy islands tend to become smaller (contain fewer molecules) and more numerous. This phenomenon is exactly within the characterization of entropic controls given in the previous section, and is, therefore, not exclusive to the Br$_2$BA molecule. As entropic control is enforced, large amorphous islands necessarily break up into smaller islands. Moreover, these small islands prefer to take on a chain shape due to the strength of the chemical control. As entropic control

is strengthened even further, chain assembly becomes increasingly repressed. This accounts for the maximum in the chain formation probability at around 240 K seen in Fig. 4b, and shows that entropic controls can be used to improve the structural quality of islands formed from the molecular self-assembly process. The analysis in the previous section also identified a symmetry breaking effect of entropic controls. This is evident in Fig. 5c,d, which shows the adsorption sites of the molecules ('colours') in a typical chain-shaped island. The disorder in the adsorption sites shows that these islands do not possess rotational symmetry despite their gross appearance in Fig. 4a. The effect of entropic controls under strong chemical control (that is, where the chemical control is large compared with $k_B T$) can, therefore, be summarized as follows: to break up amorphous islands into a number of smaller, chain-shaped islands, and to generate asymmetry in the adsorption sites of molecules within chain-shaped islands. The latter effect alters the fine structure of these islands but without significantly affecting their gross shape.

**Molecular self-assembly under intermediate chemical control.** Figure 6 presents results for the case of (NH$_2$)$_2$BA self-assembly, in which the chemical control is intermediate between Br$_2$BA case

and the Me$_2$BA case considered next. The weakened chemical control allows for amorphous island formation to occur with higher probability, however, the response of the chain-forming probability to entropic controls is essentially identical to the Br$_2$BA case. These results, therefore, highlight the fact that amorphous islands form in response to weak chemical control and not strong entropic control (large $k_BT$). Entropic control supports the formation of asymmetric islands, however, this asymmetry can be expressed through the fine structure of chain-shaped islands, and does not necessarily require the formation of amorphous islands.

**Molecular self-assembly under weak chemical control.** Figure 7 presents the case of Me$_2$BA self-assembly, in which the chemical control is the weakest of all cases studied here. In this case, amorphous island formation dominates at all temperatures studied, and the molecules assemble so that they are surrounded

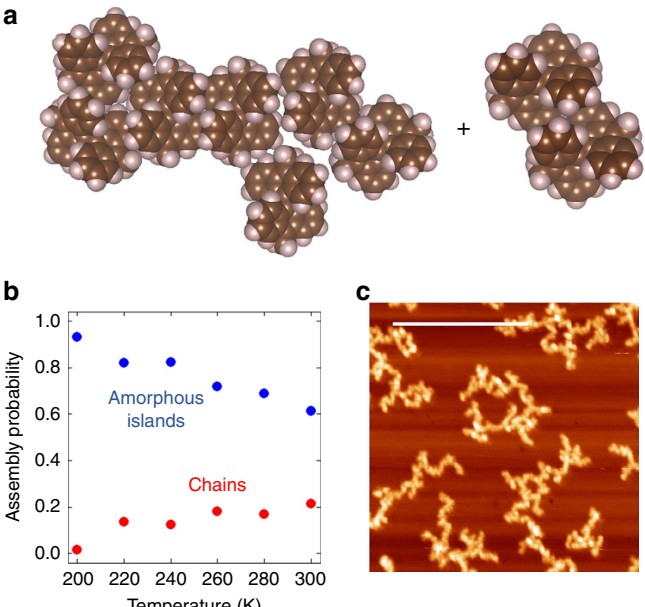

**Figure 7 | Self-assembly under weak chemical control.** (**a**) Typical low-free energy island combination formed by self-assembly of Me$_2$BA precursors on Cu(111) at 300 K. The island on the right-hand side has an amorphous shape. (**b**) Assembly probability for chain-shaped and amorphous islands. (**d**) Scanning tunnelling microscopy (STM) image of amorphous islands formed by Me$_2$BA self-assembly on Cu(111) at high temperature. The scale bar corresponds to 43 nm. STM imaging conditions: sample bias voltage = 0.1 V, tunnelling current = 10 pA, imaging temperature = 77 K, annealing temperature = 400 °C. Only amorphous islands are seen in **c**.

by other molecules. As with the previous cases, these amorphous islands typically contain several short chains due to a weak preference for the bianthryl–bianthryl interaction directionality between molecules. Enforcement of entropic controls (temperature) again breaks amorphous islands down into their constituent chains, which accounts for the increase in chain-formation probability with temperature. However, a maximum in the chain formation probability is not seen in the present calculations, and hence a very wide temperature window is necessary in order to observe all effects of entropic controls on this system. The results for the Me$_2$BA case, therefore, show that island shape becomes very unresponsive to entropic control when the chemical control is weak.

**Summary of chemical and entropic controls.** Table 1 summarizes the effects of chemical and entropic controls on the structures (islands) that emerge from the molecular self-assembly process under low surface coverage conditions, as unambiguously determined by our theoretical methodology. Chemical control affects the gross shape of the islands by increasing the likelihood of forming chain-shaped structures compared with amorphous islands. Contrary to intuition, the formation of amorphous islands was found to be a result of weak chemical control rather than strong entropic control, at the temperatures studied here. Instead, entropic control affects the fine structure of a chain-shaped islands by generating asymmetry in the molecule adsorption sites. Entropic control also degrades large amorphous islands and into large numbers of short, chain-shaped islands. The latter effect provides a method for improving the structural quality of islands *via* entropic controls. We also find that when the chemical control is weak, entropic controls become much less effective at controlling the gross island shape. While this study has characterized entropic and chemical controls in terms of chain-shaped islands, there is obvious interest for controlling the formation of other more other types of structures as well. By continued development of our code and theory, we expect to obtain increasing detailed rules for controlling molecular self-assembly and aiding the bottom-up nanomaterials fabrication process.

## Methods

**Identification of colour-shade combinations for the GBA model.** First, a realistic, symmetrized conformation for a single molecule adsorbed to the Cu(111) surface was identified using DFT calculations (Supplementary Fig. 2). Using the center of mass of the molecule as a reference point, the molecule was then scanned above the unit cell in various orientations, with the adsorption energy calculated on the fly *via* DFT (Supplementary Figs 3–5). The molecular conformation remained fixed during this process. Nine adsorption sites were identified *via* this procedure, each permitting one or two stable orientations for the molecule (Supplementary Fig. 6).

**Equivalence class sampling.** ECS was performed with in-house code written for *R* (ref. 20) using routines from various packages[22–27] (code is available upon request).

**Table 1 | Effects of chemical and entropic control on molecular self-assembly.**

|  | Weak entropic control | Intermediate entropic control | Strong entropic control |
|---|---|---|---|
| Strong chemical control | Mainly chain formation | Asymmetry in the adsorption sites of molecules in chains<br>Amorphous islands break into small chains (occurs easily) | Poor island formation |
| Intermediate chemical control | Mixture of chains and amorphous islands form | Asymmetry in the adsorption sites of molecules in chains<br>Amorphous islands break into small chains | Poor island formation |
| Weak chemical control | Mainly amorphous island formation | Asymmetry in the adsorption sites of molecules in chains<br>Amorphous islands break into small chains (does not occur easily) | Poor island formation |

'Occurs easily' and 'does not occur easily' means that the effect is easy or difficult to induce *via* entropic control (temperature), respectively.

Simulation output was visualized using VESTA[28]. The calculations reported in the following section considered 10 (initially isolated) molecules, a surface of $d = 50 \times 50 = 2,500$ unit cells (which corresponds to a surface coverage of about 3%). These calculations were ran for 600,000–1,200,000 steps of the ECS algorithm. The cut-off distance $M_c$ was set to 8 Å. The ECS algorithm was supplemented with the parallel tempering algorithm described in ref. 29. To characterize the low free-energy states of the molecular self-assembly process, 100 island combinations randomly selected from the final 300,000 steps of the simulation and used to calculate formation probabilities of chain-shaped islands and amorphous (non-chain-shaped) islands. We restrict our calculations to ten molecules and a single adsorption chirality, which is sufficient for studying chemical and entropic controls on the molecular self-assembly process in the intermediate regime. All DFT calculations were performed in VASP[30] using the rev-vdW-DF2 exchange-correlation functional[31–33]. This approach accounts for the van der Waals component of the intermolecular and molecule–surface interaction. Simulation output was visualized using VESTA[28].

Note that the cutoff distance $M_c$ should be large enough for 'weakly interacting' molecules to be present in the islands. These molecules are just close enough to the island to feel and attractive interaction, and require little energy to remove from the island. If $M_c$ is too small, then the molecules within the island will always be close together, and it will become difficult to break the island apart, resulting in longer convergence times for the Monte Carlo simulation. Moreover, 'loosely packed' islands, which might be expected to appear at high temperature, might not appear in the simulation when $M_c$ is very small. We recommend cutoff distances of at least 5 Å to reliably implement the GAMMA modelling approach.

**Data availability.** The data and codes that support the findings of this study are available upon reasonable request to the corresponding author.

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

## Acknowledgements

This research was partially supported by the World Premier Research Institute Initiative promoted by the Ministry of Education, Culture, Sports, Science and Technology of Japan (MEXT) for the Advanced Institute for Materials Research, Tohoku University, Japan. T.H. acknowledges Kakenhi No 26246022, 26108702, 26106502 and 16K14088, and support from JST (CREST). N. Asao is acknowledged for preparation of the methyl-substituted bianthryl precursor. M. Takahashi, K. Akagi and A. Staykov are thanked for computational support. The computation in this work was partially performed using the HA800-tc system RIIT at Kyushu University.

## Author contributions

D.M.P., P.H., and T.H. conceived the study. D.M.P. created the computational technique and mathematical theory, and performed all calculations. P.H. performed the scanning tunnelling microscopy experiments in T.H.'s laboratory. D.M.P., P.H. and T.H. interpreted the results. D.M.P. drafted the paper. D.M.P., P.H. and T.H. wrote the final paper together.

## Additional information

**Competing financial interests:** The authors declare no competing financial interests.

**Publisher's note**: 

