## [Peer Review File · Nature Communications]

Reviewers' comments:

Reviewer #1 (Remarks to the Author):

The manuscript of Packwood et al. presents a novel methodology for the theoretical study of the molecular self-assembly process, built on first-principles (density-functional theory) data.

The makes a clever and systematic use of first-principle calculation for the evaluation of the adsorption energy of molecules, multi-stage machine-learned empirical potential for the molecule-molecule interaction, and statistical mechanics for the implementation of the so-called generalized block assembly model.

From the application point of view, self-assembly of a class of molecule with increasing adsorption strength, the main result is correctly summarized in the abstract: entropic control (temperature) can be used to tune molecular ordering on a surface.

The manuscript presents therefore a novel methodology, applied to an interesting system and the quality of presentation is very high.

I strongly recommend this paper for publication in Nature Communication.

I have only few comments for the improvement of the manuscript:

- the GBA model maps a triangular lattice into a square lattice, with consequent change of the number of neighbours of each adsorption site with its equivalent sites. In other words, one goes from a 6-fold to a 4-fold symmetry. While the mapping of two lattice vectors, forming a plane, into a square is always geometrically legitimate, I wonder if the change of symmetry does not modify the conclusions about the self-assembly. Is it possible to design a GBA model on a triangular lattice? Would conclusions change?
- At the end of page 11, the authors state that the lack of quantitative agreement with experimental data, regarding temperature, is due to the fact that the T of the Montecarlo sampling within the GBA model is "uncalibrated". This concept becomes clear only in the Supplementary Material. Since this size dependent (number of cells D) temperature is not a usual statistical mechanics concept, it would be better if the authors can already explain it clearly in the main text.
- in the conclusions, the authors state: " Contrary to intuition, the formation of amorphous islands was found to be a result of weak chemical control rather than strong entropic control. ". I do not fully agree, in the sense that the "weakness" of the chemical control needs always to be compared to kT , which defines the scale of entropic control. So, it is always an interplay between the two aspects, one given by the specific chemistry and the other given by the temperature at which the assembly process takes place.
- for the DFT calculations, three layer of Cu(111) are used. How much is the difference in energy if 4 or 5 layers are used for few selected configurations? Is it just a constant shift when different configurations are compared?
- referring to Fig. S7, it is unclear to me how the cutoff of the dimensionality was set. Which criterion/criteria lead to the choice of 7/9/9/ principal components ?
- Fig S9, why the potential energy curve is defined as "artificial"?
- How crucial is the choice of the cut-off distance M_c ? I.e., how sensitive are results to the choice of its value?

Reviewer #2 (Remarks to the Author):

In this work, the authors develop a novel computational approach to study the self-assembly of organic molecules onto surfaces.

The approach presented here allows the authors to employ results from DFT calculations (which involve only a very small portion of the system) to obtain quantitative predictions for large-scale self-

assembly. In a sense, it is a way to obtain a coarse-grain model from DFT.

This is a relevant problem and the methodology presented here is original. As mentioned by the authors, there are other methodologies proposed in the literature with this aim of predicting thermodynamics of self-assembly, but this one is particularly original. A strong point is the possibility of obtaining results directly from DFT without the need of parameterizing force fields or potential functions. However, there are several weak points which deserve more discussion:

1.- The validity of the procedure needs to be discussed in more detail. It involves complex steps which are far from obvious. It is essential to show particular (simple) examples in which the new methodology produces equivalent results that known methodologies or reduces to known methods in the appropriate limits. In my opinion, without this validation step the methodology is unpublishable.

2.- In many places of the text it seems that the application of the method is reduced to the case of low density of adsorbed molecules. This is a very important limitation. Please clarify.

3.- The effect of the solvent is important in many self-assembly processes because solvent is present in many real applications and molecule-molecule interactions can be strongly modified by solvent (also solvent co-adsorption is possible). However, this effect seems to be impossible to be incorporated in the method. Please, comment on it.

4.- The group of Samori has devoted also a substantial effort in the development of a methodology to elucidate the entropic factors affecting self-assembly (for a review, see *Chem. Soc. Rev.*, 2012, 41, 3713-3730). The authors seem to ignore this approach. They need to take it into account.

Response to reviewer comments (NCOMS-16-14655)

We thank the reviewers for carefully reading our manuscript and providing expert feedback. Following their comments, we performed extensive density functional theory (DFT) calculations to validate the assumptions of the model. These calculations, which are described in Section 3 of the revised Supporting Information, provide strong fundamental support for the GAMMA modeling approach. Several minor revisions were also made in response to the reviewer comments. Additions to the main text are highlighted by red text.

Response to comments of Reviewer 1

- the GBA model maps a triangular lattice into a square lattice, with consequent change of the number of neighbours of each adsorption site with its equivalent sites. In other words, one goes from a 6-fold to a 4-fold symmetry. While the mapping of two lattice vectors, forming a plane, into a square is always geometrically legitimate, I wonder if the change of symmetry does not modify the conclusions about the self-assembly. Is it possible to design a GBA model on a triangular lattice? Would conclusions change?

As the reviewer correctly points out, the GBA model reduces the symmetry of the lattice from 6-fold to 4-fold symmetry. However, this will not affect the conclusions about self-assembly, for the following reasons.

Consider the energetic component of our calculations (equation (1) of the main paper). Because every unique adsorption site is contained in the parallelogram formed by the two lattice vectors, the surface-molecule interaction term in our force field has the correct range of values. The interaction energy component is also not affected by the reduction in lattice symmetry: given two molecules (two cell-color-orientation combinations in the model), the interaction energy is calculated after first mapping back to the real coordinate space where the atoms lie. Thus, the energy of any configuration of the GBA model is equivalent to the corresponding energy in the real coordinate space. Note that the atomistic images of the islands shown in Figures 4, 6, and 7 were also drawn by mapping the GBA model configurations back to this real coordinate space.

Now consider the entropic component of our calculations (equation (3) of the main paper). The entropic component is determined by the degeneracy factor $n(q)$ for each island

combination q (see equation (3)). The conclusions about molecular self assembly, as described at the end of the ‘Methods’ section will not be affected if $n(q)$ was calculated using other unit cell shapes. This is because the explicit shape of the unit cell does not enter into the calculation of $n(q)$. In this calculation, each unit cell is condensed into a single point, and the calculation of $n(q)$ is performed with respect to a point lattice (see Figure S15). In particular, $n(q)$ is calculated by considering the number of ways of choosing distinct sets of points from this lattice, and does not directly rely on the exact symmetry of the lattice.

In principle, there is no restriction on the type of lattice for the GBA model, providing that at least one of each unique adsorption site is contained in the unit cell and that the case where two molecules adsorbed at the same unit cell can be ignored. For best use of the GBA model, we recommend considering large molecules and small unit cells. We make brief mention of the above points in a new paragraph at the end of the ‘Methods’ section of the revised manuscript.

At the end of page 11, the authors state that the lack of quantitative agreement with experimental data, regarding temperature, is due to the fact that the T of the Montecarlo sampling within the GBA model is "uncalibrated". This concept becomes clear only in the Supplementary Material. Since this size dependent (number of cells D) temperature is not a usual statistical mechanics concept, it would be better if the authors can already explain it clearly in the main text.

We have moved our explanation of this point from the Supporting Information to a new paragraph at the end of the ‘Methods section’.

- in the conclusions, the authors state: " Contrary to intuition, the formation of amorphous islands was found to be a result of weak chemical control rather than strong entropic control. ". I do not fully agree, in the sense that the "weakness" of the chemical control needs always to be compared to kT , which defines the scale of entropic control. So, it is always an interplay between the two aspects, one given by the specific chemistry and the other given by the temperature at which the assembly process takes place.

The reviewer correctly points out that the ‘weakness’ of the chemical control should always be compared to kT . We briefly mention this in the ‘Results and Discussion’ section.

- for the DFT calculations, three layer of Cu(111) are used. How much is the difference in energy if 4 or 5 layers are used for few selected configurations? Is it just a constant shift when different configurations are compared?

In Section 3 of the revised Supporting Information, the island-surface interaction energy is computed for several small islands on a 3-layer Cu(111) and a 4-layer Cu(111) surface. The use of a 4-layer surface reduces this interaction energy by a nearly constant amount of about -0.16 eV with a variation of about 0.01 to 0.08 eV. This variation is small compared to the sizes of the total energies of the islands.

However, note that the island-surface interaction energy is actually approximated by the sum of the molecule-surface interaction energies in our force field. In order to validate the use of a 3-layer Cu(111) surface in calculating the force field parameters, we therefore computed the adsorption energy map for a single Br₂BA molecule on the unit cell of a 4-layer Cu(111) slab, and found no significant qualitative or quantitative differences from the 3-layer calculations. This strongly supports the use of a 3-layer surface in the calculation of the molecule-surface interaction energy terms in the GBA model. Please see section 3 of the Supporting Information for further discussion. In addition, other calculations in section 3 justify using the sum of molecule-surface interaction energies to approximate the island-surface interaction energies.

- referring to Fig. S7, it is unclear to me how the cutoff of the dimensionality was set. Which criterion/criteria lead to the choice of 7/9/9/ principal components ?

In principal component analysis, the number of principal components (cut-off of the dimensionality) is typically chosen such that around 90 % the total variation in the data is accounted for. In Figure S10 (previously S7), the first 7 principal components for the Br₂BA case account for 94.6 % of the total variation in the data, and so 7 was chosen as the number of principal components. In the (NH₂)₂BA and Me₂BA cases, the first 9 principal components account for 93 % and 93.5 % of the total variation in the data, respectively, and so 9 principal components were chosen for these cases. We now explicitly mention these points in the caption of Figure S10. In addition, Figure S10 now plots the cumulative variance accounted for by the principal components.

- Fig S9, why the potential energy curve is defined as "artificial"?

In Figure S9 (S12 of the revised SI), the blue Lennard-Jones (LJ)-type curve is a generic image that was not calculated directly. In this sense, the curve is ‘artificial’. For improved clarity, we have now removed the confusing word ‘artificial’. The blue curve was actually adapted from a (non-copyrighted) image available online, and a reference to the source is given in the revised SI.

- How crucial is the choice of the cut-off distance M_c ? I.e., how sensitive are results to the choice of its value?

The cut-off distance is the maximum distance that a single molecule within an island may be separated from the other molecules in the island. If this distance is exceeded, then this molecule becomes its own island. It is therefore important that the cut-off distance is large enough to allow for ‘weakly interacting’ molecules to appear in the islands. These molecules are close enough to the island to feel an attractive interaction, but also far enough away that they could depart with a small thermal fluctuation. Whenever a weakly interacting molecule appears in an island in our Monte Carlo calculations, this molecule becomes likely to detach from the island due to the entropy pay-off of creating a new island. If the cut-off distance is too small, then the molecules within the island will always be close together and will become difficult to separate from the island. This will increase the equilibration time of the Monte Carlo simulation, and moreover may fail to predict ‘loosely-packed’ islands that might appear at high temperature. We recommend cut-off distances that exceed about 5 Å in order to perform these calculations accurately.

In our calculations, weakly interacting molecules were never observed to be separated from the rest of the island by more than about 4 Å, which is well within the cut-off distance 8 Å used in our calculations. In one of the 100 island combinations randomly sampled for the high-temperature (300 K) Br₂BA simulation, we observed one molecule separated from one of the islands by 4.76 Å. Such a weak interaction occurs with a probability of less than 1 % at high temperature. We briefly mention these points in a new paragraph at the end of the ‘methods’ section.

Response to comments of Reviewer 2

1.- The validity of the procedure needs to be discussed in more detail. It involves complex steps which are far from obvious. It is essential to show particular (simple) examples in which the new methodology produces equivalent results that known methodologies or reduces to known methods in the appropriate limits. In my opinion, without this validation step the methodology is unpublishable.

In response to this important comment, a large number of DFT calculations were performed to test the assumptions of our theoretical model (section 3 of the Supporting Information). In our opinion, a rigorous examination of the model's assumptions is the most meaningful way to validate our methodology.

Let us first note that, prior to the introduction of the force field in equation (1), the GBA model makes three assumptions: (i) that each unit cell can only hold one adsorbed molecule at a time; (ii) that only a discrete set of adsorption sites exist in each unit cell; (iii) that only a discrete set of orientations is possible for the adsorbed molecules. (i) is justified for all but the smallest of molecules, providing that the surface unit cell is not too large. (ii) and (iii) are justified for the present system by the adsorption energy maps calculated in Figures S3 – S6, which show extensive corrugation in the surface-molecule interaction energy. (iii) is further justified by the widely-known fact that bianthracene molecules align with the three atomic row directions of the (111) surface. Therefore, there is no reason to doubt physical picture of the GBA model, providing that there is strong epitaxial interactions between the molecule and underlying substrate.

Before discussing the force field of the GBA, let us consider the degeneracy factor $n(q)$ used to calculate the entropy in equation (3). Our calculation of this quantity is accurate under low-coverage conditions, as shown in the proof in section 5.5 of the Supporting Information, and does not invoke any other physical assumptions. Note that a very similar proof was subject to peer-reviewed when we published a paper on equivalence class sampling theory for a mathematical audience (*Roy. Soc. Open. Sci.* **3**, 2016, 150681). The entropic component of the free energy is therefore accurately calculated by our Monte Carlo calculations, providing low-coverage conditions apply.

The validity of the GBA model under low-coverage conditions is therefore determined entirely by the force field in equation (1). The force field makes three assumptions: (1)

that the total island-surface interaction energy can be approximated by a sum of molecule-surface interaction energies, (2) that the total intermolecular interaction energy can be approximated by the sum of pairwise interaction energies, and (3) that the surface-molecule interaction energies and pairwise molecule-molecule interaction energies are not affected by the other molecules in the island.

To validate Assumption (1), we selected several small islands from our simulations and compared the actual island-surface interaction energy with the sum of the molecule-surface interaction energies (Table S3). The comparison was quite good, with the sum of the molecule-surface interactions overestimating the island-surface interaction by only 3 – 4 %. These results therefore support the use of Assumption (1). This overestimation of the island-surface interaction energy is probably due to the formation of a small ‘charge cloud’ between the molecule and surface, which would have a minor destabilizing effect on the island-surface interaction. Discussion about this point is presented at the end of Section 3.1. Further investigation into this matter is an interesting chemical physics problem, but is beyond the scope of the paper.

To validate Assumption (2), we compared the total intermolecular interaction energy of these islands with the sum of the pairwise interaction energies (Table S4). The comparison here was again very good, with the pairwise interactions underestimating or overestimating the total interaction energy by around 3 %. This calculation strongly suggests that higher-order contributions to the intermolecular interaction can be neglected in the energy calculation.

Assumption (3) is equivalent to saying that the state of an adsorbed molecule is not affected by the presence of other molecules. ‘State’ refers to the adsorption location, orientation, and conformation of the adsorbed molecule. To test this assumption, we attempted DFT relaxation on three representative islands that appeared from our Monte Carlo simulations. As Figure S8 shows, the molecules in these islands hardly moved during the DFT relaxation process. This suggests that the islands predicted by our method indeed occupy positions of local energy minima in the potential energy landscape. However, we cannot rule out the possibility that these islands might adopt a different configuration over a very long time-scale due to the presence of other nearby local minima.

In passing, note that the machine learning method used to calculate the pairwise

interaction energies is already validated extensively in Section 4 of the Supporting Information. The mathematical theory that guarantees the accuracy of these methods is thoroughly documented. For example, see M. Mohri, A. Rostamizadeh, and A. Talwalkar. *Foundations of Machine Learning* (MIT, 2012).

In summary, our approach is rigorously validated from both mathematical and physical viewpoints. It is particularly important to note that Assumptions (1 – 3) discussed above would be present in virtually any coarse-grained self-assembly model. In this sense, the GBA model is equivalent to many other coarse-grained self-assembly models. The differences with our approach are the use of the low-coverage approximation to provide an analytical calculation of the entropy, and the use of machine learning to estimate the pairwise interaction energies, both of which are well-justified. We make brief mention of these points at the end of the ‘Methods’ section of the main paper.

2.- In many places of the text it seems that the application of the method is reduced to the case of low density of adsorbed molecules. This is a very important limitation. Please clarify.

By restricting ourselves to the low-coverage conditions, we can obtain an analytical calculation of the entropy (actually, degeneracy) for a particular combination of islands on the surface. In turn, this allows us to unambiguously separate the effects of chemical and entropic control on the self-assembly process. The low-coverage approximation is obviously a limitation if one is wanting a general molecular self-assembly theory, however it is advantageous for acquiring thermodynamic insights into the self-assembly process. In particular, other theoretical methods cannot obtain such insights directly *via* an analytic formula.

It may be possible to calculate the degeneracy factors under general conditions. If this can be achieved, then the methodology presented here would immediately become applicable away from low-coverage conditions. These points are now emphasized in a new paragraph at the end of the ‘Methods’ section.

3.- The effect of the solvent is important in many self-assembly processes because solvent is present in many real applications and molecule-molecule interactions can be strongly modified by solvent (also solvent co-adsorption is possible). However, this effect seems to be impossible to be incorporated in the method. Please, comment on it.

The reviewer correctly points out that solvents have a major effect on the self-assembly process, particularly due to the coordination of the solvent to the molecules and islands. It indeed does not seem possible to incorporate solvent effects into our method, and we briefly point this out at the start of the ‘Methods’ section.

4.- The group of Samori has devoted also a substantial effort in the development of a methodology to elucidate the entropic factors affecting self-assembly (for a review, see Chem. Soc. Rev., 2012, 41, 3713-3730). The authors seem to ignore this approach. They need to take it into account.

Unfortunately, we were unaware of the work Samori *et al.* A brief discussion of Samori *et al.*'s conclusions on entropy for the self-assembly process is made in a new paragraph at the end of the ‘Methods’ section.

It is interesting to compare Samori *et al.*'s conclusions with our results. In addition to emphasizing the importance of low final enthalpy in the self-assembled structure (which corresponds to strong chemical control), they emphasize that small and ‘soft’ self-assembled structures have the higher thermodynamic likelihood of occurring. ‘Soft’ structures are those with low-frequency vibrational modes. The connection between small structures and entropy corresponds to our finding that entropy favors the formation of a large number of small islands. We suspect that there is a connection between the dynamical concept of ‘soft’ islands and the statistical mechanical concepts of distinguishable and low symmetry islands as discussed in our paper. Deducing this connection is a deep chemical physics problem that goes beyond the scope of our paper, however we mention it at the end of the third section of the main paper.

Other changes

The adsorption energy maps for Br₂BA on the Cu(111) unit cell (Figure S3 of the Supporting Information) was plotted with the incorrect color scale in the previous manuscript. This problem has now been fixed.

Reviewers' comments:

Reviewer #1 (Remarks to the Author):

I am fully satisfied with the replies given by the authors to my questions and remarks, and with the consequent changes in the manuscript.

I therefore strongly recommend this paper for publication in Nature Communications.

Reviewer #2 (Remarks to the Author):

Report of Referee

In this revised version, the authors have modified substantially their manuscript. They have added many clarifications and performed additional calculations in order to validate their approach. The revised version presents more clearly the approach and the objectives: a coarse-grain methodology fed from quantum mechanical DFT calculations designed to elucidate the role of entropy in self-assembly of molecules over surfaces. It is important to note that the entropy is calculated analytically from equations valid only at low densities, so the method is aimed at the study of low density situations as stated clearly in this revised version of the manuscript.

The method seems promising but I am still unconvinced by the response of the authors to two of my previous objections. Here are my points:

1) Lack of proper validation of the method. The authors have included a substantial amount of new DFT calculations in order to prove the assumptions of the method involved in Eq. (1). But I was not questioning Eq.(1). I agree with the authors that their calculations show that the hypothesis behind Eq.(1) are valid (basically they prove that the different interactions can be well approximated by additive pairwise interactions). But there are other essential hypotheses in the model which are more difficult to prove that are assumed without further proof. These are mainly (a) the use of a Ising-type model for describing adsorbed molecules (b) the use of lattice model with a symmetry imposed "a priori" and (c) the use of a Monte Carlo simulation method with a nonphysical temperature T ("uncalibrated T", as the authors wrote). The justification of these hypotheses is difficult and the only way I see to test it is by comparison with standard approaches in simplified models (as I suggested in my previous report). I suggest performing simulations of simplified models solvable by other, standard techniques and comparing the results and the validity of hypotheses (a), (b),(c) . The use of grids during the simulation is particularly dangerous (because introduces non-physical symmetries), and for this reason other methods aiming at the study of self-assembly (for example Ref 7) use grid interpolation in order to avoid explicit use of grids in the Monte Carlo simulations.

2) The aim of the model is to be able to evaluate the entropic contribution to the self-assembly process and clarify the role of chemical control (enthalpy) and entropy in self-assembly over surfaces. For this reason, the authors limit to low coverage situations. As far as I understand it, this is the same aim as in the methodology that I mentioned in my previous report (now Ref 21: Chem. Soc. Rev., 2012, 41, 3713-3730) but in that case the calculations are not limited to low density. In addition, the prediction of the present model is only qualitative because the temperature reported in Figs 4, 7 is not the real physical temperature (it is "uncalibrated" according to the text, whatever this means). Therefore, the classification of entropic and enthalpic (chemical) control presented in Table 1 is qualitative, in contrast with the equivalent predictions of the method in Ref 21 which are quantitative. Without a proper definition of temperature we cannot evaluate the magnitude of the entropic contribution and compare it quantitatively with enthalpy (chemical control). I think that these drawbacks can be corrected without having to do again all the work (the DFT calculations and the fits to Eq.1 are fine). I think that the methods for entropy calculation worked out in ref 21 can be adapted easily to the present method, so the limitation of low density can be overcome. Also the use of an

"uncalibrated" temperature in Monte Carlo can be corrected by working out the statistical mechanics of the problem, I do not see why the authors use such a meaningless concept.

We thank the reviewers for carefully considering our revised manuscript and providing their comments. Following these comments, it became clear that some parts of our manuscript were vague and subject to misinterpretation. We have therefore made the following alternations to our manuscript to ensure proper comprehension.

- Clarified explanation of ‘temperature’ in our theory

We have emphasized that the **temperature in our theory is the usual temperature from thermodynamics** (end of the first paragraph of the Results section) . We have also emphasized that the temperature range over which our calculations were performed differs from that of the experiments because we consider a very low density of molecules on the surface (see the end of the first paragraph of the Results section). In the previous manuscript we used the term ‘uncalibrated temperature’ to summarize this point, however this created the false impression that our temperature is not the usual thermodynamic temperature. The term ‘uncalibrated temperature’ has now been deleted.

- Additional evidence to justify the relevance of the low-coverage approximation

In the Supporting Information (section S5.6), we provide some mathematical evidence that the concepts yielded by our theory (which is restricted to low surface coverages of molecules) are relevant even at higher surface coverages. We show that under any surface coverage, $S = S_0 + S_c$, where S is the actual entropy, S_0 is the low-surface-coverage entropy (which we calculate exactly), and S_c is a correction term (which is unknown, but extremely small under low surface coverage conditions). Thus, under any surface coverage condition, our theory describes the effect of the term S_0 in the entropy expression. Especially under low surface coverage conditions, our theory describes the effect of entropy S ($\approx S_0$) with high accuracy.

- Further technical explanation of our model

We provide minor additional emphasis at several points (see the red text in our manuscript), as described in our response to Referee 2 below.

Referee 2 makes several comments which we disagree with. Our response is as follows.

The revised version presents more clearly the approach and the objectives: a coarse-

grain methodology feed from quantum mechanical DFT calculations designed to elucidate the role of entropy in self-assembly of molecules over surfaces. It is important to note that the entropy is calculated analytically from equations valid only at low densities, so the method is aimed at the study of low density situations as stated clearly in this revised version of the manuscript.

It is also important to note that low density situations are the only situations where the equations for the entropy can be straightforwardly interpreted. This allows us to rigorously deduce new concepts regarding the effect of entropy on the molecular self-assembly process, which is the main merit of this work.

The method seems promising but I am still unconvinced by the response of the authors to two of my previous objections. Here are my points:

1) Lack of proper validation of the method. The authors have included a substantial amount of new DFT calculations in order to prove the assumptions of the method involved in Eq. (1). But I was not questioning Eq.(1). I agree with the authors that their calculations show that the hypothesis behind Eq.(1) are valid (basically they prove that the different interactions can be well approximated by additive pairwise interactions). But there are other essential hypotheses in the model which are more difficult to prove that are assumed without further proof. These are mainly (a) the use of a Ising-type model for describing adsorbed molecules (b) the use of lattice model with a symmetry imposed “a priori” and (c) the use of a Monte Carlo simulation method with a nonphysical temperature T . The justification of these hypotheses is difficult and the only way I see to test it is by comparison with standard approaches in simplified models (as I suggested in my previous report). I suggest performing simulations of simplified models solvable by other, standard techniques and comparing the results and the validity of hypotheses (a), (b),(c) . The use of grids during the simulation is particularly dangerous (because introduces non-physical symmetries), and for this reason other methods aiming at the study of self-assembly (for example Ref 7) use grid interpolation in order to avoid explicit use of grids in the Monte Carlo simulations.

Our methodology has been rigorously justified. We respond to points (a – c) in turn.

(a) the use of a Ising-type model for describing adsorbed molecules

Point (a) is vague and apparently contradicts the reviewer’s own admission that they agree

with Equation (1) (which defines an Ising model). The main assumption involved behind an Ising model (or equivalently, equation (1)) is that the molecules only adsorb at particular points on the surface and in particular orientations. These restrictions are fully justified by the DFT calculations in the Supporting Information (Figures S3 – S5), which show extensive corrugation in the molecule-surface interaction potential and an overwhelmingly strong surface-molecule interaction energy. The molecules therefore tend to adsorb at particular locations and align with the lattice planes of the surface. Incidentally, these assumptions are also behind the ‘surface-assisted molecular self-assembly mechanism’, which was proposed before our study and is widely accepted to apply to bianthracene molecules adsorbed to Cu(111) (*ACS Nano* **8**, 2014, 9181; *ACS Nano* **9**, 2015, 12035). Evidentially, our model does not hold when surface-assisted molecular self-assembly does not take place, and we now mention this point in the manuscript (see the text beneath equation 1).

(b) the use of lattice model with a symmetry imposed “a priori”... The use of grids during the simulation is particularly dangerous (because introduces non-physical symmetries), and for this reason other methods aiming at the study of self-assembly (for example Ref 7) use grid interpolation in order to avoid explicit use of grids in the Monte Carlo simulations.

We responded to this exact point when addressing the comments of Reviewer 1 during the first round of review. Reviewer 1 was very much satisfied with our response. As mentioned then, the apparent 4-fold symmetry of the GBA model in Figure 2B is never imposed during the Monte Carlo simulation, for the following reasons.

- The image in Figure 2B is a shorthand notation for the *real space* configuration of the molecules on the surface. Given a cell, color and shade (a molecule), we map back to the real 3D space where the surface and molecule atoms lie, and perform our calculations there. The ‘grid’ on which our calculations are performed consists of the actual adsorption sites on the surface (see the newly added bottom panel of Figure S6 of the Supporting Information and below). **The symmetries of this grid, as well as the adsorption energies for molecules sitting at these points, are determined by the surface-molecule interaction potential.** As far as I understand, mapping the square-grid configurations of the GBA model to the real space adsorption sites will actually have the same effect as the ‘grid interpolation’ method mentioned by the reviewer.

Figure S6 (bottom) (newly added to the Supporting Information). Grid of adsorption sites generated from 25 of the unit cells in Figure S6 (top panel). A single unit cell is indicated by the red parallelepiped. All configurations of the GBA model are mapped to this grid when performing energy calculations.

- Given two molecules (two cell-color-orientation combinations in the GBA model), the interaction energy is also calculated after first mapping back to the real coordinate space where the atoms lie. Thus, **the energy of any configuration of the GBA model is equivalent to the corresponding energy in the real coordinate space.**
- The entropic component is determined by the degeneracy factor $n(q)$ for each island combination q (see equation (3)). In the calculation of $n(q)$, each unit cell is condensed into a single point, and then $n(q)$ is calculated by considering the number of ways of choosing distinct sets of points from this lattice. **The symmetry of the lattice does not enter into the entropy calculation.**

Thus, **the apparent four-fold symmetry of the GBA model is never imposed during the calculations**; all calculations are performed on a grid formed by the real adsorption sites of the molecule, and whose symmetry is determined by the real surface-molecule interaction potential (we recall our previous comment that, in the case of surface-assisted molecular self-assembly, surface-molecule interaction potential alone determines where the molecules sit on the surface). We had added a few sentences to explain this point (see the second-to-last paragraph of the Method section).

(c) the use of a Monte Carlo simulation method with a nonphysical temperature T

The temperature in our model is the exact thermodynamic temperature. However, we used the inappropriate terminology ‘uncalibrated temperature’ in our manuscript, which created the false impression that our temperature is non-physical. As mentioned previously, we have removed this expression from our manuscript.

Our Monte Carlo method assumes a very low density of molecules on the surface. This means that a very low thermodynamic temperature is needed to stabilize island formation. At higher temperatures, the molecules will spend little time interacting and most of their time migrating across the surface (recall that the molecule-molecule interaction is very weak). They will very rarely collide or coalesce, and any islands that do form will quickly fall apart and will infrequently appear in the Monte Carlo sampling. The experiments discussed in Figure 4 and 7 are performed at higher surface coverages. This means that, even at higher temperatures, the migrating molecules will frequently collide/coalesce to form islands. While the molecules in these islands can quickly escape, new molecules can quickly join. Thus, island formation is observed under higher temperature conditions in the experiments than that in our simulations. The restriction to low coverages is extremely useful for gaining conceptual insights into molecular self-assembly, as we describe soon.

As for the Monte Carlo simulation method (‘Equivalence Class Sampling’) itself, a **rigorous mathematical validation of this methodology** was recently published by us (reference 20). Reference 20 formally proves that Equivalence Class Sampling becomes increasingly exact as the size of the surface increases with the number of molecules held fixed. This fact provides the ultimate justification of our simulation method, and renders unnecessary the need to validate against ‘*standard approaches in [sic] simplified models*’ (as suggested by the reviewer). If one wishes to disagree, then disproval of the theorems in Reference 20 and in section 5 of the Supporting Information of our current manuscript will be necessary.

2) The aim of the model is to be able to evaluate the entropic contribution to the self-assembly process and clarify the role of chemical control (enthalpy) and entropy in self-assembly over surfaces. For this reason, the authors limit to low coverage situations. As far as I understand it, this is the same aim as in the methodology that I mentioned in my previous report (now Ref 21: Chem. Soc. Rev., 2012, 41, 3713-3730) but in that case the calculations are not limited to low density.... I think that

these drawbacks can be corrected without having to do again all the work (the DFT calculations and the fits to Eq.1 are fine). I think that the methods for entropy calculation worked out in ref 21 can be adapted easily to the present method, so the limitation of low density can be overcome.

There is a major difference between reference 21 and our current paper. While the method presented in reference 21 apparently calculates a numeric value for the entropy, it provides little *conceptual understanding* of the effect of entropy on molecular self-assembly. The complicated formulas from reference 21 require extensive numerical computation and cannot be unambiguously interpreted. The reviewer stresses that the theory in reference 21 applies under more general coverage conditions than our theory. However that very generality limits the conceptual insights the theory affords. As a case in point, reference 21 introduces a concept called ‘structural softness’ to describe the effect of entropy on the self-assembly process. However this concept is never formally defined and its connection with the equations is never rigorously proven.

The methodology presented in our paper addresses all of the above problems: our compact, analytical formulas (5) and (6) unambiguously characterize the effect of entropy on the self-assembly process in terms of island rotational and translational symmetry. The concepts presented in our paper are not mentioned in reference 21, and would be very difficult to deduce from the theory presented there. As an applied mathematician, I am adamant that the purpose of theory is to *rigorously deduce concepts* rather than calculate numbers.

On the other hand, we appreciate that the ‘best’ approach to dealing with entropy is a matter of opinion and differs from person to person. There may be readers who prefer the entropy calculation in reference 21 to our formulas (5) and (6), and for their benefit we emphasize that our methodology can be easily modified to incorporate other formulas for the entropy (third-to-last paragraph of the Methods section).

In addition, the prediction of the present model is only qualitative because the temperature reported in Figs 4, 7 is not the real physical temperature (it is “uncalibrated” according to the text, whatever this means). Therefore, the classification of entropic and enthalpic (chemical) control presented in Table 1 is qualitative, in contrast with the equivalent predictions of the method in Ref 21 which are quantitative. Without a proper definition of temperature we cannot evaluate the

magnitude of the entropic contribution and compare it quantitatively with enthalpy (chemical control)....

Following our previous comments, the temperature in our model is the real physical temperature, and the classification of entropy and chemical control is quantitative (Table 1 and the discussion are presented qualitatively. This is preferred for communicating our results to a general audience).

Also the use of an “uncalibrated” temperature in Monte Carlo can be corrected by working out the statistical mechanics of the problem, I do not see why the authors use such a meaningless concept.

The implication here is that the low-density restrictions of our theory can be corrected by simply ‘*working out the statistical mechanics of the problem*’. Such a calculation would require an analytic computation of the number of ways that n arbitrary-shaped tiles can be placed on a square lattice, which is mathematically impossible unless a very low density of tiles is assumed. Such a calculation may be possible *via* numerical analysis and a lot of patience. However, in such case the theory would be of limited conceptual value.

REVIEWERS' COMMENTS:

Reviewer #2 (Remarks to the Author):

I am satisfied by the detailed responses that the authors provided to my comments. I think that the paper has been substantially improved during all the discussions. The method proposed here is extremely promising. I think it is time to publish it so that the community at large will be able to employ it and discuss about its performance and results.